# Stachydrine, a Bioactive Equilibrist for Synephrine, Identified from Four *Citrus* Chinese Herbs

**DOI:** 10.3390/molecules28093813

**Published:** 2023-04-29

**Authors:** Yifei Sun, Xuexue Xia, Ganjun Yuan, Tongke Zhang, Beibei Deng, Xinyu Feng, Qixuan Wang

**Affiliations:** 1Biotechnological Engineering Center for Pharmaceutical Research and Development, Jiangxi Agricultural University, Nanchang 330045, China; 2Laboratory of Natural Medicine and Microbiological Drug, College of Bioscience and Bioengineering, Jiangxi Agricultural University, Nanchang 330045, China

**Keywords:** choline, *Citrus*, method, analysis, decoction, Aurantii Fructus Immaturus, Aurantii Fructus, effect, cardiovascular protection, uterus

## Abstract

Four Chinese herbs from the *Citrus* genus, namely Aurantii Fructus Immaturus (*Zhishi*), Aurantii Fructus (*Zhiqiao*), Citri Reticulatae Pericarpium Viride (*Qingpi*) and Citri Reticulatae Pericarpium (*Chenpi*), are widely used for treating various cardiovascular and gastrointestinal diseases. Many ingredients have already been identified from these herbs, and their various bioactivities provide some interpretations for the pharmacological functions of these herbs. However, the complex functions of these herbs imply undisclosed cholinergic activity. To discover some ingredients with cholinergic activity and further clarify possible reasons for the complex pharmacological functions presented by these herbs, depending on the extended structure–activity relationships of cholinergic and anti-cholinergic agents, a simple method was established here for quickly discovering possible choline analogs using a specific TLC method, and then stachydrine and choline were first identified from these *Citrus* herb decoctions based on their NMR and HRMS data. After this, two TLC scanning (TLCS) methods were first established for the quantitative analyses of stachydrine and choline, and the contents of the two ingredients and synephrine in 39 samples were determined using the valid TLCS and HPLC methods, respectively. The results showed that the contents of stachydrine (3.04‰) were 2.4 times greater than those of synephrine (1.25‰) in *Zhiqiao* and about one-third to two-thirds of those of *Zhishi*, *Qingpi* and *Chenpi*. Simultaneously, the contents of stachydrine, choline and synephrine in these herbs present similar decreasing trends with the delay of harvest time; e.g., those of stachydrine decrease from 5.16‰ (*Zhishi*) to 3.04‰ (*Zhike*) and from 1.98‰ (*Qingpi*) to 1.68‰ (*Chenpi*). Differently, the contents of synephrine decrease the fastest, while those of stachydrine decrease the slowest. Based on these results, compared with the pharmacological activities and pharmacokinetics reported for stachydrine and synephrine, it is indicated that stachydrine can be considered as a bioactive equilibrist for synephrine, especially in the cardio-cerebrovascular protection from these citrus herbs. Additionally, the results confirmed that stachydrine plays an important role in the pharmacological functions of these citrus herbs, especially in dual-directionally regulating the uterus, and in various beneficial effects on the cardio-cerebrovascular system, kidneys and liver.

## 1. Introduction

The fruits or peels of some citrus genus plants were traditionally used as *Qi*-regulating Chinese herbs [1]. Among them, the herbs Aurantii Fructus Immaturus (*Zhishi*) and Aurantii Fructus (*Zhiqiao*) are collected from the dried young and near-mature fruits, respectively, of *Citrus aurantium* L., *Citrus sinensis* Osbeck or their cultivated varieties, and harvested from May to June and in July, respectively. These herbs are also widely used in prescriptions for treating various gastrointestinal and cardiovascular diseases in clinics, and many ingredients, including alkaloids, flavonoids, essential oil, coumarins, limonoids, etc., were identified and considered to be responsible for their pharmacological functions [2,3].

It was reported that two alkaloids, synephrine and *N*-methyltyramine, discovered in these herbs can produce various adrenergic effects including raising blood pressure, constricting peripheral blood vessels, dilating the pupil, stimulating the uterus and relaxing the intestines [3,4,5]. However, some animal experiments and clinical results indicated that the decoctions or aqueous extracts of *Zhishi* and *Zhiqiao* connoted some pharmacological effects contrary to synephrine and *N*-methyltyramine [6]. For example, (1) transiently or inapparently raising, or sometimes even lowering blood pressure [7,8]; (2) various cardiovascular protections [9,10]; (3) an excitatory effect to the in vitro and in vivo uteruses of pregnant and nonpregnant rabbits [11]; (4) a two-way regulating effect on gastrointestinal smooth muscle, not only exciting the gastrointestinal tract and enhancing its peristalsis, but also reducing the tension of gastrointestinal smooth muscle and relieving spasm [12]. These facts showed that there are probably some other ingredients with cholinergic or anti-cholinergic activities in the decoctions or aqueous extracts of *Zhishi* and *Zhiqiao*, although some reports indicated that citrus flavonoids and essential oil should play a certain role in these complicated functions [13,14,15].

Based on the above, it is easy to associate these ingredients with the agonists of cholinergic receptors or the antagonists of adrenergic receptors, the former being more probable. The extended structure–activity relationships indicated whether cholinergic or anticholinergic agents present a similar structural characterization (Figure 1). As shown in the center of Figure 1, a nitrogen atom of a tertiary amine or quaternary ammonium is linked to an oxygen atom by an alkane chain of 2–3 carbons, or by a hydrocarbon chain of 3–4 carbons containing *cis*-double bonds, with the commensurate space distance. This structural fragment can bind with the cholinergic receptor or choline esterase, and then initiate various cholinergic or anticholinergic activities. This led us to focus on the discovery of choline analogs from these herb decoctions. As choline does not contain any chromophore and has no ultraviolet absorption, it is unable to probe according to our previous structure-oriented thinking using the HPLC method [16]. However, choline has the structural fragment of quaternary ammonium, and belongs to alkaloids. Thus, an efficient and specific chromogenic reagent was first explored here for establishing a sensitive thin-layer chromatography (TLC) method to detect choline analogs from these citrus herbs or other plant resources. Then, probable choline analogs in these herbs were isolated and identified, and the quantitative analyses of them and synephrine were performed for clarifying the substance bases of various pharmacological activities and presenting possible reasons for the complex pharmacological functions presented by these Chinese citrus herbs.

## 2. Results

### 2.1. Detection of Choline Analogs in These Chinese Citrus Herbs

For detecting possible choline analogs from botanical resources, choline chloride was used as a positive indicator. Considering that synephrine, *γ*-aminobutyric acid and their analogs were already discovered in these four Chinese citrus herbs [2,3], *γ*-aminobutyric acid and synephrine were taken as two controls to exclude the similar primary and secondary amines, respectively, in many botanical samples. Various proportions of two solvent systems I (ethyl acetate–95% ethanol–formic acid) and II (*n*-butanol–glacial acetic acid–water) were tested for botanical samples, and the results indicated that there was no obvious difference in developing effects between these two solvent systems. The better proportions of solvent systems I and II were 10:4:5 and 4:1:5 (lower layer), respectively. Considering that solvent system II developed more slowly on the plate than solvent system I, and that a tiering operation should be simultaneously completed for solvent system II, solvent system I with the proportions of 10:4:5 was selected for the subsequent TLC analyses.

Furthermore, the color reactions (Appendix A) on the thin-layer plates indicated that choline and synephrine presented brown and orange-red strips, respectively, for improved Dragendorff reagents, while no strips were observed from all the lanes of choline, synephrine and *γ*-aminobutyric acid after being colored with Dragendorff and Wagner reagents. Although Dragendorff and Wagner reagents are two broad-spectrum chromogenic agents for alkaloids, they generally do not work for many compounds with lower molecular weights, such as choline and synephrine. From Appendix A, it is indicated that the improved Dragendorff reagent can effectively detect possible choline analogs, which usually have lower molecular weights. It can also give synephrine, a catecholamine amine with a low molecular weight, an orange-red color. This further indicated that a compound presenting a red or orange-red color, like many alkaloids, is unlikely a choline analog after being colored with the improved Dragendorff reagent.

According to the general procedure of TLC analysis, possible choline analogs in four Chinese citrus herbs were detected with a positive indicator of choline, using the optimized developing solvent and chromogenic reagent. The results (Figure 2a) showed that all four herbs contained choline or a probable choline analog. There are probably more analogs and larger contents in the decoction sample of *Zhishi* than those of *Zhiqiao*, Citri Reticulatae Pericarpium Viride (*Qingpi*) and Citri Reticulatae Pericarpium (*Chenpi*). The TLC analyses further indicated that samples from different producing areas contain different choline analogs with different contents, such as the TLC analyses for ten *Zhishi* samples from different producing areas (Figure 2b).

### 2.2. Isolation and Identification of Choline Analogs in Citrus Chinese Herb Zhishi

Considering that the decoctions of *Zhishi* samples contained more choline analogs and higher contents, ten samples of *Zhishi* from different producing areas (Figure 2b) were mixed and pulverized for obtaining more probable analogs of choline. Using the preparative TLC, probable choline analogs **1**, **2** and **3** were isolated from the sampling solution of *Zhishi* according to the process in Section 4.2.2. Based on their spectroscopic data of HRMS, or ^1^H and ^13^C NMR, compounds **1**, **2** and **3** were identified as choline, stachydrine (Figure 3) and synephrine, respectively. This indicated that there is a similar fragment, which is surrounded by a blue coil for stachydrine in Figure 3, in the structure of stachydrine and choline.

The related data for their identification are as follows.

**1**: A white amorphous powder; a brown spot colored with the improved Dragendorff reagent on the thin-layer plates; HRESIMS *m/z* 104.1060 [M]^+^ (calcd. for C_5_H_14_NO, 104.1075); ^1^H NMR (MeOH-*d*_4_, 600 MHz) *δ*: 3.91 (H-3), 3.56 (H-2) and 3.24 (H-4 to H-6, s); ^13^C NMR (MeOH-*d*_4_, 150 MHz) *δ*: 68.9 (C-2), 56.9 (C-3) and 54.5 (C-4, 5 and 6). These NMR data are in accordance with those reported for choline [17].

**2**: A white amorphous powder; a reddish-brown spot colored with the improved Dragendorff reagent on the thin-layer plates; HRESIMS *m/z* 144.1010 [M]^+^ (calcd. for C_7_H_14_NO_2_, 144.1025); ^1^H NMR (D_2_O, 400 MHz) *δ*: 4.07 (1H, t, *J* = 9.2 Hz, H-2), 3.70 (1H, m, H-5a), 3.53 (1H, m, H-5b), 3.29 (3H, s, H-6), 3.10 (3H, s, H-7), 2.48 (1H, m, H-3a), 2.33~2.24 (1H, m, H-3b) and 2.23~2.11 (2H, m, H-4); ^13^C NMR (D_2_O, 100 MHz) *δ*: 171.3 (C-8), 76.4 (C-2), 67.2 (C-5), 52.1 (C-6), 45.9 (C-7), 25.3 (C-3) and 18.6 (C-4). These NMR data are in accordance with those reported for stachydrine [18].

**3**: A white amorphous powder; an orange-red spot colored with the improved Dragendorff reagent on the thin-layer plates; HRESIMS *m/z* 168.1009 [M + H]^+^ (calcd. for C_9_H_14_NO_2_, 168.1024), 150.0903 [M + H − H_2_O]^+^ (calcd. for C_9_H_12_NO, 150.0919) which presented as base peak ion.

The above indicated that compound **3** is not a choline analog. After spraying with the improved Dragendorff reagent, the spots of compounds **1** and **2**, two choline analogs, presented brown or reddish-brown color, while compound **3** showed an orange-red spot on the thin-layer plates. These indicated that the compounds represented by the spots are unlikely choline analogs if the color of the spots is red. A detailed discussion is given in Section 3.

### 2.3. The Contents of Stachydrine, Choline and Synephrine in Four Chinese Citrus Herbs

Along with the identification of stachydrine and choline from the four Chinese citrus herbs Zhishi, Zhiqiao, Qingpi and Chenpi, it is known that three alkaloids, namely stachydrine, choline and synephrine, are extensively distributed in these herbs, and their proportions and contents in these herbs are different from each other. As we mentioned in the introduction, the decoctions or aqueous extracts of Zhishi and Zhiqiao connoted probable cholinergic activities, contrary to the adrenergic effects of synephrine. Simultaneously, these herbs have different pharmacological functions according to the theory of Chinese medicine. Thus, it was worth exploring whether there were some relationships between the complex pharmacological functions of these herbs and the contents and proportions of these three ingredients. To achieve this, the contents of these three ingredients were determined.

#### 2.3.1. Validation of Quantitative Analyses

As there is no chromophore in the structures of stachydrine and choline, it is unsuitable to perform their quantitative analyses using the HPLC-UV method. Based on the detection procedure of choline analogs and their separation effects on thin-layer plates, the TLC scanning (TLCS) method was considered for the quantitative analyses of stachydrine and choline. Meanwhile, the quantitative analysis for synephrine was established using the HPLC-UV method, referring to that described in the Chinese pharmacopeia [1].

The methodology validation showed that the chromatographic peaks of both choline and stachydrine presented good symmetry, and both compounds can be well separated with an identical resolution of 1.26 from their nearest peaks, according to the procedure of TLCS analysis described in Section 4.4.1. The limit of detection (LOD) and limit of quantitation (LOQ) for stachydrine were 1.0 μg (0.20 mg·mL^−1^) with a relative standard deviation (RSD) of 0.08% and 2.0 μg (0.40 mg·mL^−1^) with an RSD of 0.18%, respectively, and those for choline were 0.5 μg (0.10 mg·mL^−1^) with an RSD of 0.08% and 0.8 μg (0.15 mg·mL^−1^) with an RSD of 0.18%, respectively. The repeatability tests showed that the RSDs for a strip of stachydrine and choline on a thin-layer plate were 0.34% and 0.48%, respectively. The precision tests indicated that the RSDs for six strips of stachydrine and choline on the same thin-layer plate were 1.20% and 3.55%, respectively, and those for the identical solutions of stachydrine and choline on six thin-layer plates were 4.05% and 4.15%, respectively. A good linearity correlation y = 6565.7x + 5522.7 (r = 0.9996) (Appendix A) between the amounts (x) and peak areas (y) was presented for stachydrine in a range from 2.0 to 14.0 μg, and an acceptable linearity correlation y = 12302.1x − 909.3 (r = 0.9987) (Appendix A) was presented for choline in a range from 1.0 to 4.0 μg. Using a powder sample of Zhishi (No. 2010001), the reproducibility was assessed, and the results showed that the contents of stachydrine and choline in this sample were 0.302% (with an RSD value of 2.39%) and 0.085% (with an RSD value of 2.03%), respectively. Using this Zhishi sample, the recoveries of choline and stachydrine were tested. The results showed that the average recovery of stachydrine was 100.28% with an RSD value of 3.08 (Appendix A), and that of choline was 101.04% with an RSD value of 2.31 (Appendix A). These results together indicated that the established TLCS methods were valid, and can be used for the quantitative analyses of stachydrine and choline in these herbs.

Similarly, the chromatographic peaks of synephrine presented good symmetry and can be well separated with a resolution of 4.06 from its nearest peaks, according to the procedure of HPLC analysis described in Section 4.5.1. The limit of detection (LOD) and limit of quantitation (LOQ) for synephrine were 0.02 μg and 0.05 μg, respectively. The repeatability experiments presented an RSD value of 0.42% for a standard solution of synephrine. A good linearity correlation y = 4435448.7x + 2217.2 (r = 1.0000) (Appendix A) between the amounts (x) and peak areas (y) was presented for synephrine in a range from 0.5 to 32.0 μg. Using a powder sample of Zhishi (No. 2010001), the reproducibility experiment showed that the contents of synephrine in this sample were 1.10% with an RSD value of 1.77%, and the sample solution showed good stability with an RSD value of 1.60% in 24 h. Using this Zhishi sample, the recovery of synephrine was tested, and the results showed that the average recovery of synephrine was 99.89% with an RSD value of 3.19 (Appendix A). These results together indicated that the established HPLC method was validated and can be used for the quantitative analyses of synephrine in these herbs.

#### 2.3.2. Contents of Stachydrine, Choline and Synephrine in Four Chinese Citrus Herbs

Using the above validated methods for the quantitative analyses, the contents of stachydrine, choline and synephrine in these citrus herbs were determined, and the results are shown in Table 1.

From Table 1, the contents of the three ingredients in the four herbs fluctuate greatly. Overall, the contents of stachydrine and synephrine are obviously higher than those of choline, and the content fluctuations of synephrine are greater than those of stachydrine in these herbs. Simultaneously, the contents of synephrine and stachydrine are obviously higher in Zhishi than in the other three herbs. It is worth noting that the content of synephrine in the herb Zhiqiao is the lowest among these herbs, while that of stachydrine in the herb Zhiqiao is very high and just lower than that in the herb Zhishi.

#### 2.3.3. Statistical Analysis for the Content Data of Three Ingredients

To make the content differences between these three ingredients clearer to explain the differences in the pharmacological functions of these herbs, the data in Table 1 were further analyzed using statistical methods. The results confirmed that the contents of stachydrine and synephrine in the herb Zhishi are larger than in the other three herbs (*p* < 0.05 or *p* < 0.01), and those of choline in Qingpi (or/and Zhishi) are larger than in Zhiqiao and Chenpi (*p* < 0.05 or *p* < 0.01). Simultaneously, stachydrine is the largest of these three ingredients in Zhiqiao (*p* < 0.05 or *p* < 0.01).

Since herbs Zhishi and Zhiqiao are the dried young fruits of *Citrus aurantium* L. (or its cultivated varieties), and Qingpi and Chenpi are the dried peel from young (or immature) fruits of Citrus reticulata Blanco (or its cultivated varieties), it was further concluded that all three ingredients will reduce (*p* < 0.05) with the prolongation of growth time for Citrus genus plants, except for stachydrine in the herbs Qingpi and Chenpi. Among them, the contents of synephrine decrease the most rapidly, while those of stachydrine decrease the most slowly. Although all their contents reduce with the prolongation of growth time for these Citrus genus plants, the contents of synephrine are significantly higher than those of stachydrine and choline, except those of stachydrine are largest in the herb Zhiqiao. Moreover, since herbs Zhishi (or Zhiqiao) and Qingpi (or Chenpi) are derived from the fruits and the pericarps, the fact that synephrine and stachydrine distribute more in the exocarp than in the mesocarp could be further inferred from their decreasing speeds and degrees with the growth time of these Citrus genus plants, and this inference was also in accordance with a previous report [19]. However, there is possibly no obvious difference in the distribution of choline in exocarp and mesocarp.

### 2.4. Comprehensive Analyses for the Pharmacological Effects of Stachydrine and Synephrine

These four herbs are derived from the fruits or peels of *Citrus* genus plants. Herbs *Zhishi* and *Zhiqiao* originate from the fruits of *Citrus aurantium* L. or its cultured varieties at different harvest times, and the herbs *Qingpi* and *Chenpi* originate from the peels of *Citrus reticulata* Blanco or its cultured varieties at different harvest times [1]. Their decoctions or water extracts have various pharmacological effects on the digestive system, cardiovascular system, respiratory system and so on [2,3,11]. Many reports indicate that flavonoids, alkaloids, coumarins, essential oil and limonoids are the main components of these herbs [2,3]. It was reported that some flavonoids (such as narirutin, naringin, hesperidin, neohesperidin and nobiletin) and alkaloids (such as synephrine and *N*-methyl tyramine), with higher content and a wider distribution in these herbs, have various bioactivities in the digestive system, cardiovascular system and respiratory system, and which are mainly responsible for the pharmacological effects of these herbs [11,12,20]. However, some pharmacological activities of these herbs in the human body have few related investigations, such as the excitatory effect of the herbs *Zhishi*, *Zhiqiao* and *Qingpi* on the in vitro and in vivo uteruses of both pregnant and non-pregnant rabbits. Moreover, some related investigations remain insufficiently clear, such as which components are responsible for the two-way regulating effects on gastrointestinal smooth muscle, showed by the decoction of *Zhiqiao* [12,20].

Considering that the decoctions or aqueous extracts of these herbs conceal some possible cholinergic activities, here, probable choline analogs in these herbs were detected and determined. The results from Table 2 show that stachydrine (or plus choline) and synephrine presented commensurate contents in these herbs and similar changing trends with the increase in growth time. However, the contents of synephrine decrease most rapidly, while those of stachydrine decrease most slowly. Since synephrine is a sympathomimetic amine and has intrinsic sympathomimetic activity, it was inferred that they possibly have different or even contrary pharmacological activities. If this is true, different contents and proportions of the two compounds in these herbs would have important impacts on their pharmacological functions in the human body, which would probably give some reasonable interpretations for the difference in pharmacological functions of these herbs, although other components also have important roles. Moreover, this would also fluctuate the pharmacological effects of these herbs with different producing areas and harvest times. To clarify these, the main bioactivities of stachydrine (plus choline) and synephrine were summarized and are listed in Table 3.

From Table 3, synephrine, as a partial agonist of *α*_1_-adrenoreceptor and an antagonist of *α*_2_-adrenoreceptor, can constrict peripheral blood vessels, cerebrovascular and aorta, while stachydrine, an ingredient coexisting with synephrine in these citrus herbs, presents various cardio-cerebrovascular protections including rapid vascular relaxation, accelerating blood circulation, increasing coronary and myocardial blood flow, relieving myocardial necrosis, slowing heart rate and decreasing cardiac output, suppressing and ameliorating myocardial fibrosis, ameliorating cardiac hypertrophy and fibrosis. Some results were obtained from animal models induced by adrenergic receptor agonists (marked in bold font in Table 3). Simultaneously, synephrine would increase the level of platelet [37], while stachydrine can inhibit platelet aggregation and ameliorate platelet-mediated thrombo-inflammation [27,28,38]. Synephrine can contract the uterus (pregnancy) [47], while stachydrine can regulate the uterus, such as through the inhibition of convulsive uterus, the stimulation of uterine contraction [49] and reducing uterine bleeding [52]. Moreover, synephrine can be rapidly absorbed and predominantly metabolized in the liver [72], and this would lead to some possible unfavorable influences on the liver, especially at large doses of citrus herbs or some related juices containing synephrine. However, stachydrine can rapidly relax blood vessels by activating the endothelial nitric oxide synthase in the vascular endothelium [35], and has various helpful effects on the liver, such as anti-inflammatory action, ameliorating hepatic fibrosis [44] and treating non-alcoholic fatty liver [26]. Thus, the unfavorable influence of synephrine on the liver would be theoretically eradicated by stachydrine coexisting in these citrus herbs.

From the pharmacokinetic characteristics (Table 3) of synephrine and stachydrine, both compounds can be rapidly absorbed after oral administration. The relative bioavailability is approximately 22% for synephrine and 90% for stachydrine, respectively. Considering that the contents of stachydrine in these herbs except *Zhiqiao* are approximately one-third to two-thirds of that of synephrine, some pharmacological contributions from the bioavailable differences in the two ingredients would be balanced, to a great extent, by their contents in these herbs. Together with most contrary pharmacological activities mentioned above (shown in Table 3), these indicate that both ingredients can be considered, only from their pharmacological activities, as a pair of antagonists in these citrus herbs. However, they both present many contrary bioactivities and also show some different or synergetic bioactivities, such as neuroprotective, hepatoprotective, renal protective, antitussive and anti-inflammatory effects from stachydrine, and gastrointestinal relaxation, antidepressant and anti-obesity activities from synephrine, along with their synergistic activities in uterine contraction and anti-diabetes. Thus, when they are used for other medicinal purposes, their contrary bioactivities, usually acting as their individual adverse effects, can be partly canceled out by each other. From this view, they can also be considered as a pair of synergists or associates from their contributions to the pharmacological functions and safety of these herbs.

It is noteworthy that there are no related reports for another compound with some of the pharmacological activities that are presented in Table 3 for synephrine or stachydrine. However, this does not mean that it has no similar, different or even contrary bioactivities to the identical organ or tissue. As described above, stachydrine was eventually identified from these herbs, using the detection method of choline analogs. Simultaneously, from the pharmacological activities, especially various cardio-cerebrovascular effects, stachydrine presents various contrary effects compared to synephrine and other adrenergic receptor agonists in Table 3 (marked in bold font for items (1), (5), (6) and (8) of the row “cardio-cerebrovascular system”). Thus, it is indicated that stachydrine seems to have some cholinergic activities, and is more like an agonist of M-type cholinergic receptor. Considering that their contents changed with the harvest time of these herbs simultaneously and similarly, stachydrine and synephrine can be also considered as a pair of bioactive equilibrists in the *Citrus* genus, like a pair of sympathetic and parasympathetic neurotransmitters in the human body.

Based on the above, many convoluted pharmacological functions for the aqueous extract or the decoction of *Zhishi*, *Zhiqiao*, *Qingpi* or/and *Chenpi* can be scientifically and rationally interpreted, including an excitatory effect on the in vitro and in vivo uteruses of both pregnant and non-pregnant rabbits, reducing cerebrovascular resistance and increasing cerebral blood flow, constricting gallbladder and a two-way regulating effect on gastrointestinal smooth muscle. Moreover, some explanations for the four Chinese herbs harvested from the different parts and growth times of the *Citrus* genus having different functions are presented in Section 3.

## 3. Discussion

Inspired by the traditional and modern pharmacodynamics of four Chinese citrus herbs, here, possible choline analogs were discovered from these herbs using the TLC method with a specific chromogenic reagent, which led to the identification of stachydrine and choline based on their NMR and HRMS data. After this, a TLCS method was first established for the quantitative analyses of stachydrine and choline, and the contents of both ingredients and synephrine in 39 samples were determined. Based on this, the statistical analyses of the contents of these three ingredients were performed, and then the pharmacological effects and pharmacokinetics reported for stachydrine and synephrine were comprehensively compared and analyzed. The results showed that stachydrine and synephrine can be considered as a pair of bioactive equilibrists, especially in the cardio-cerebrovascular protection from these citrus herbs, and which can, to a great extent, present some reasonable interpretation for the complex pharmacological functions of these herbs. Moreover, some important and relevant aspects are further discussed and developed as follows.

### 3.1. A Simple Method Detecting Choline Analogs from Plant Resource

Based on the extended structure–activity relationships of cholinergic and anti-cholinergic agents (Figure 1), choline analogs were speculated to be agonists or antagonists of cholinergic receptors, and have various bioactivities in the cardio-cerebrovascular, digestive and nervous systems, and the eyes, among others. Simultaneously, some of them also belong to betaine analogs, and these compounds have various bioactivities including anti-ulcer, regulating gastrointestinal function and treating liver diseases, and have multiple effects on homocysteine metabolism, which is very helpful for the protection of the cardio-cerebrovascular system and the kidneys [73]. For example, stachydrine was recently reported to have very extensive bioactivities (Table 3), and it was also isolated from another Chinese herb *Yimucao* (the overground part of *Leonurus japonicus* Houtt.) [26]. Based on the above TLC procedure for discovering stachydrine and choline from these citrus herbs, a simple and rapid method was established for quickly discovering possible choline analogs from botanical resource, and schemed as Figure 4.

The procedure (Figure 4) can be used for the discovery of choline analogs, including some betaine analogs. There are two key factors for discovering choline analogs: one is the specific chromogenic reagent (improved Dragendorff’s reagent), and the other is the color of the spots. Generally, two chromogenic reagents, Dragendorff’s and Wagner, are used for the color reaction of most alkaloids, but they present poor effects for some alkaloids with a low molecular weight. Considering that an acidic environment should be provided for the color reaction, the acetic acid was replaced with phosphoric acid in Dragendorff’s reagent for improving the chromogenic sensitivity, referring to what Zhang N, et al. reported [74]. Moreover, if the color of the spots is red, such as orange-red or brownish-red, the compounds represented by the spots are likely not choline analogs but rather alkaloids, possibly some multi-methoxy flavonoids, or multi-methoxybenzenes (such as *α*- and *β*-asarones), since the potassium bismuth iodide reagent colors these compounds to orange-red or brownish-red. This was also supported by the fact that compound **3** was found not to be a choline analog, and colored an orange-red spot **c** (Figure 3). Furthermore, the brown color can be more accurately defined, as the absorption wavelengths ranged from 550 to 580 nm when the thin-layer plate was scanned in 20 to 50 min after being taken from the chromogenic reagent.

Moreover, possible choline analogs can be quickly identified by the combinational method of preparative TLC and ^1^H NMR, since these compounds would present one to three single peaks with some specific ^1^H chemical shifts ranging from 3.20 to 3.60 ppm (4.45 to 4.75 ppm for methylpyridine-type choline analogs, such as trigonelline), assigned to three to nine hydrogens (one to three methyl groups). It is noteworthy that these analyses should eliminate the signal peaks from the deuterated methanol used for the NMR experiments or the possible residual methanol in the process of sample preparation. Thus, the deuterated solvents MeOH-*d*_4_ should be avoided to use for the NMR experiments as much as possible. Considering that these compounds contain a structural fragment of the quaternary ammonium, D_2_O is considered the preferred solvent.

According to the LODs of stachydrine and choline, the detectable content of choline analogs in botanical resources is approximately 1.0 to 1.4 μM per gram of sample powder, and equal to contents of 0.01% to 0.05% in dried botanical resources, assuming the molecular weights of choline analogs are less than 500. If a highly efficient thin-layer plate is used for the method, the detection sensitivity would be increased. Using this detection procedure of choline analogs, choline and stachydrine were also detected in samples from the herbs *Xiangyuan* (dried fruits of *Citrus wilsonii* Tanaka) and *Foushou* (dried fruits of *Citrus medica* L. var. *sarcodactylis* Swingle) (Appendix A), and this was also in accordance with previous publications [43,75]. These indicated that choline and stachydrine are widely distributed in Chinese herbs of the *Citrus* genus. Moreover, the results showed that stachydrine and/or choline were also discovered in the leaves of *Citrus* genus plants, with mostly higher contents than in their individual fruits (Appendix A). Moreover, some possible choline analogs can be also detected in other botanical resources, including some Chinese herbs such as *Huangliang* (Coptidis Rhizoma), *Juhua* (dried capitulum of *Chrysanthemum morifolium* Ramat.) and *Chuanxiong* (dried rhizome of *Ligusticum chuanxiong* Hort.) (Appendix A). This indicates that the method can effectively detect choline or its analogs in various botanical resources.

### 3.2. The Contents of Stachydrine, Choline and Synephrine in These Citrus Herbs

According to the theory of Chinese medicine, these four citrus herbs have different pharmacological functions [1]. To clarify their substance bases and the reasons for their different functions, many studies on their active ingredients such as flavonoids, alkaloids and essential oil were reported [2,3,76]. Citrus flavonoids and alkaloids were generally considered the two main components responsible for the pharmacological functions of these herbs [76]. However, it remains a convoluted fact that for all the pharmacological activities of the ingredients reported in these herbs, it is difficult to clarify the pharmacological functions of their decoctions. Among them, synephrine acts as an agonist of adrenoreceptor, and is widely distributed in these herbs. Here, stachydrine and choline were also discovered from these herbs and were reported to have various bioactivities. Considering that stachydrine and synephrine can be considered as a pair of bioactive equilibrists, it was expected to give some reasonable explanations for the convoluted pharmacological functions presented by these herbs. Thus, the contents of these three ingredients in these herbs were further determined.

As there is no chromophore in the structures of stachydrine and choline, it is unsuitable to perform their quantitative analyses using the HPLC-UV method, although this method was used for the quantitative analyses of stachydrine in Yimucao and choline in various plants. According to the detection procedure of choline analogs, stachydrine and choline can be perfectly isolated from their adjacent spots in these herbs (Figure 1). Thus, here, the TLCS method was selected for the quantitative analyses of stachydrine and choline, although the HPLC-MS/MS method can also be used for both components [71].

The quantitative analyses indicated these three ingredients have similar changing trends with the increase in growth time. The contents of all three compounds decrease from *Zhishi* (harvested in June) to *Zhiqiao* (harvested in July), and from *Qingpi* (harvested in July) to *Chenpi* (harvested in January of next year). However, the decreasing speed of stachydrine is slower than that of synephrine, which was also supported by the depth analyses for the ratio values of stachydrine and synephrine, comparing *Zhishi* (0.44 ± 0.16) with *Zhiqiao* (2.43 ± 1.03) (*p* < 0.05) and *Qingpi* (0.34 ± 0.14) with *Chenpi* (0.67 ± 0.33) (*p* < 0.05). Although they were collected from different plants, the changing trends of synephrine, choline and stachydrine in 39 batches of samples are certain according to statistical analyses (Table 2), and among them, those of synephrine in various citrus herbs were also supported by many reports [76].

It is noteworthy that some multi-methoxy flavonoids having various bioactivities in the cardiovascular system, such as nobiletin and tangeretin [77], are reported to be widely distributed in the fruits or peels of many *Citrus* genus plants, including these herbs [78,79]. Simultaneously, their contents in the peels are much higher than those in other tissues (such as sarcocarp and seed) of these herbs [78,80], in accordance with the distribution of stachydrine in the fruits and peels of these herbs. Moreover, their contents in the fruits or peels of *Citrus* genus plants also present a decreasing trend with the delay of harvest time [81,82], which is similar to the decreasing trend of stachydrine and choline. It was reported that stachydrine is a proline betaine, and choline is also a precursor of glycine betaine [73,83,84]. All the bio-syntheses of stachydrine, choline and multi-methoxy flavonoids were catalyzed by *S*-adenosyl-methionine (SAM)-dependent methyltransferases with a universal methyl donor SAM [85,86,87]. Differently, the sub-classified *N*-methyltransferases are responsible for the bio-syntheses of stachydrine and choline, while the sub-classified *O*-methyltransferases are responsible for that of multi-methoxy flavonoids [87]. The similar changing trends of stachydrine/choline and multi-methoxy flavonoids in the same tissue of the *Citrus* genus plants indicate there is a kind of intrinsic mechanism for simultaneously regulating both sub-classified enzymes with a similar effect. More probably, there are some physiological needs regulating the whole biosynthesis pathway involving SAM-dependent methyltransferases down in the fruits and peels of these citrus genus plants. Thus, it is worth further studying the physiological regulations of these *Citrus* genus plants to these components in their fruits and peels, which would be very helpful for clarifying the different pharmacological functions of these herbs and the regulation relationship of the two sub-classified enzymes in *Citrus* genus plants.

### 3.3. Communication between Active Ingredients and Pharmacological Effects of These Herbs

It was reported that these citrus herbs mainly contain flavonoids, alkaloids, essential oils and coumarins [2,15,88]. Many experiments have indicated that *Citrus* flavonoids exert multiple beneficial effects on cardiovascular and metabolic health through antioxidant, antidiabetic and anti-inflammatory activities, and by modulating lipid metabolism and adipocyte differentiation, etc. [89,90]. Simultaneously, essential oils have extensive pharmacological activities in the central nervous system, such as sedation, hypnosis, anti-anxiety and anti-depression, and also in the digestive system, including gastro- and hepato-protective effects [91]. Considering that the decoctions of these herbs were usually used for treating some diseases, the contributions of these ingredients to the pharmacological effects from the application of these herbs would be lower than anticipated because of the weak hydrophilicity of many flavonoids, essential oil and coumarins in these herbs. Here, two water-soluble components, stachydrine and choline, present extensive biological activities, including beneficial effects on the cardio-cerebrovascular and nervous systems and the kidney, liver, and blood, and obviously regulating uterus effects (pregnancy and non-pregnancy), among others. (Table 3). Taken together with the contents of stachydrine and synephrine in these herbs, and the balance from their pharmacokinetics after oral administration, it was inferred that this pair of equilibrists plays an unignorable role in the pharmacological effects of these herb decoctions.

Some reports indicated that aqueous extracts or the decoctions of these herbs can contract the in vitro and in vivo uteruses of both pregnant and non-pregnant rabbits, and relax the in vitro uterus of pregnant rats or mice [92,93,94]. Ahangarpour et al. confirmed that the aqueous extract of *C. aurantium* flowers can reduce spontaneous motility and decrease the uterus contractions of pregnant rats, related to voltage-dependent calcium channels and without involving *β*-adrenoceptors and opioid receptors [95]. From Table 3, many reports have indicated that stachydrine can not only stimulate uterine contraction but also inhibit uterine spasms in both pregnant and non-pregnant rabbits [27,48,49,51]. Thus, the discovery of stachydrine with good water solubility and considerable content in these herbs can provide some reasonable interpretations for the heterogeneous effects of these herb decoctions on the uterus, depending on the different contents of stachydrine and other related components in these herbs and the physiological states of the uterus. Moreover, choline has similar effects on the uterus as stachydrine [51], while its contents in these herbs are only about 1/5 to 1/25 of those of stachydrine (Table 2).

Traditionally, these herbs are used for the treatment of some cardiovascular disease, and have various beneficial effects on cardiovascular health, including antioxidant and anti-inflammatory, hypolipidemic, anti-thrombosis and anti-atherosclerosis effects, and cardio-cerebrovascular protection, etc. [3,96,97]. Recently, Mahmoud et al. summarized that *Citrus* flavonoids confer cardiovascular protection via their antioxidant, antidiabetic, anti-inflammatory, anti-atherosclerosis and other biological activities [89,90], and it was reported that some citrus flavonoids such as a multi-methoxy flavonoid nobiletin have anti-hypertensive activity [6]. Simultaneously, Pontifex et al. also pointed out that *Citrus* fruits should be encouraged in the diet for their potential neurological benefits [98]. However, they also recommended that further studies and clinical trials should be performed for evaluating their efficacy. Simultaneously, it is noteworthy that the poor bioavailability of many citrus flavonoids dampens their systemic effects after oral administration, due to the combination of their degradation by intestinal bacterial enzymes, the poor hydrophobic nature of aglycone, the efflux of intestinal P-glycoprotein and the metabolism of cytochrome P450 [99,100,101]. Moreover, the concentrated distributions of some citrus flavonoids such as tangeretin and naringenin in the kidney, lung and liver will also influence the actual effects on the cerebrovascular system [102,103]. Thus, these factors would reduce the actual effects of citrus flavonoids on the cardio-cerebrovascular system, although their beneficial effects are confident. Based on these findings, stachydrine with high bioavailability (above 90%) and good tissue distribution would play an important role in cardiovascular and cerebral protection, since it has extensive beneficial effects on the cardio-cerebrovascular system (Table 3). Moreover, the rapid vascular relaxation, slowing heart rate and decreasing cardiac output, and the increasing coronary and myocardial blood thanks to stachydrine can also offset the possible adverse effects of synephrine and *N*-methyltyramine on the heart for non-drug purposes, which is probably one of the reasons why blood pressure sometimes presented an inapparent or transient raise and even decline after the oral administration of these herb decoctions to experimental animals [7,8]. Furthermore, Liu et al. reported that choline also has anti-hypertensive and cardiovascular protective effects [104].

It was reported that the concentrated solutions (1.5 g dried herbs per milliliter) of these citrus herb decoctions have obvious in vitro antiplatelet aggregation activities in human platelets [105]. Synephrine would increase the level of platelets [37]. Conversely, stachydrine can not only inhibit platelet aggregation but also ameliorate platelet-mediated thrombo-inflammation [27,28,38]. Thus, considering that stachydrine has good water solubility and bioavailability, and its considerable contents in these herbs, it can be confirmed that stachydrine is an important component in the anti-platelet aggregation of these herb decoctions, although some flavonoids, such as hesperidin and naringin [106,107], were also reported to have inhibitory activity against platelet aggregation [89,90].

Furthermore, choline is already used for the treatment of nonalcoholic fatty liver disease [46,108]. Thus, it can be deduced that choline in the decoctions of these herbs, especially *Qingpi*, would play an important role in maintaining the function and health of the liver [45,46]. Moreover, choline is also beneficial for cardiovascular and atherosclerosis diseases, and possibly neurological disorders [109,110]. Besides the protective effects of *Citrus* flavonoids on the cardiovascular system mentioned above, they also have beneficial effects on gastrointestinal health and function [13,14], and renal, hepatic and nervous system protections [111,112,113]. Nevertheless, the two-way regulating effects of these herb decoctions on gastrointestinal smooth muscle should result from the combinational effects of flavonoids, essential oil and synephrine [13,15,92]. Simultaneously, citrus flavonoids, coumarins, stachydrine and choline should play main roles in the neuroprotective effects of these citrus herbs [2,88,113], while essential oil, *γ*-aminobutyric acid and synephrine should be responsible for various functions of the central nervous system, such as sedation, hypnosis, anxiolytic, anticonvulsant and anti-depression [11]. Moreover, citrus flavonoids, essential oil and stachydrine in these herbs could have antitussive and expectorant effects on the respiratory system.

Taken together, stachydrine should play an important role in the pharmacological functions of these citrus herbs, especially it can dual-directionally regulate the uterus and has various beneficial effects on the cardio-cerebrovascular system, blood, kidney and liver. Simultaneously, as a pair of bioactive equilibrists, the cardio-cerebrovascular protection of stachydrine can counteract the possible cardiovascular risk brought out from synephrine, which is very beneficial for the safe use of these citrus herbs. Moreover, together with the pharmacological activities of alkaloids (choline, synephrine, *N*-methyl tyramine and *γ*-aminobutyric acid) and *Citrus* flavonoids, essential oil and coumarins, these can more scientifically and reasonably interpret the substance bases for the various pharmacological effects of these citrus herb decoctions. Conversely, the differences in the formation, content, water solubility, extractability and pharmacokinetic characteristics of these components would lead to efficient differences among these four Chinese citrus herbs.

## 4. Materials and Methods

### 4.1. Materials, Chemicals and Reagents

Ten dried samples (Table 1) of *Zhishi*, *Qingpi* and *Chenpi*, together with nine samples (Table 1) of *Zhiqiao*, from different places of production were purchased from Chengdu Huichu Technology Co., Ltd. (Chengdu, China). Moreover, eight leaf and fruit samples of *Citrus aurantium* L. and its cultivated varieties, *Citrus junos* Siebold ex Tanaka, and *Citrus reticulata* Blanco ‘Zhangtouhong’ were collected from different habitats and growth ages and were identified by senior agronomist Yao Nie, and some fresh leaves and fruits of *Xiangyuan* were collected in Xinyu, China. The following Chinese herbs were purchased from Yifeng Pharmacy (Xialuo Branch, Nanchang, China): *Foshou*, the dried fruits of *Citrus medica* L. *var*. *sarcodactylis* Swingle (Guangxi, China); *Xiangyuan*, the dried fruits of *Citrus wilsonii* Tanaka; *Huanglian*, the dried rhizome of *Coptis chinensis* Franch (Sichuan, China); *Chuanxiong*, the dried rhizome of *Ligusticum chuanxiong* Hort. (Sichuan, China); *Dafupi*, the dried peel of *Areca catechu* L. (Yunnan, China); *Banxia*, the processed products according to the legal process for the dried tuber *Pinellia ternata* (Thunb.) Breit; *Juhua*, the dried capitulum of *Chrysanthemum morifolium* Ramat.; *Mahuang*, or Ephedrae Herba; *Duzhong*, the dried bark of *Eucommia ulmoides* Oliv.; *Kushen*, the dried root of *Sophora flavescens* Ait.; and *Gancao*, or Glycyrrhizae Radix et Rhizoma.

Reference standard choline chloride (No. C12799084) with a purity of 98% was purchased from Shanghai Macklin Biochemical Co., Ltd. (Shanghai, China), and stachydrine hydrochloride (No. PS012344) and synephrine (No. PS000966), each with purity of more than 98%, were purchased from Chengdu Push Bio-Technology Co., Ltd. (Chengdu, China).

TLC silica gel plates were purchased from Qingdao Ocean Chemical Co., Ltd. (Qingdao, China), and HPTLC Silica gel 60 F254 was purchased from Merck KGaA (Darmstadt, Germany). Bismuth subnitrate used for preparing the chromogenic reagent of TLC analyses was purchased from Aladdin Chemistry Co., Ltd. (Shanghai, China). Chromatographic-grade methanol (Anhui Tedia High Purity Solvents Co., Ltd., Anqing, China) and sodium dodecyl sulfonate (Shanghai Baihe Chemical Plant, Shanghai, China) were used for the HPLC analyses of synephrine. The ultra-pure water was prepared by the TST-UPB-10 ultra-pure water machine (Shijiazhuang TST Equipment Co., Ltd., Shijiazhuang, China). Other chemicals (analytical purity) were purchased from China National Pharmaceutical Group Co., Ltd. (Beijing, China).

### 4.2. Detection of Choline Analogs

#### 4.2.1. Controls and Chromogenic Reagents

Choline chloride was used as a positive control, and *γ*-aminobutyric acid and synephrine were taken as excluded controls. Two chromogenic reagents (Dragendorff’s and Wagner) and an improved Dragendorff reagent for alkaloids were selected for color reactions on the thin-layer plate. The stock solutions were prepared by the equal mixture of solution I (0.85 g of bismuth nitrite dissolved in 10.0 mL of acetic acid with analytical purity and 40.0 mL of water) and solution II (8.00 g of potassium iodide dissolved in 20.0 mL of water), and 1.0 mL of the stock solution, 2.0 mL of acetic acid (analytical reagent) and 10.0 mL of water were mixed as Dragendorff’s reagent before use. Then, 1 g of iodine and 10.0 g of potassium iodide were dissolved in 50 mL of water, and the Wagner reagent was prepared by transferring this solution into a 100 mL volumetric flask, subsequently supplementing it with 2.0 mL of acetic acid and the required amount of water to make a constant volume. The improved chromogenic reagent was prepared by adding a mixture of bismuth nitrite (0.82 g), potassium iodide (11.06 g) and 50% (*v*/*v*) phosphoric acid (90 mL) into a 100 mL volumetric flask to make a constant volume using water.

#### 4.2.2. Reference and Sample Solutions

A total of 25.0 mg each of choline chloride, stachydrine hydrochloride and synephrine was transferred into a volumetric flask, and then a concentration of 1.0 mg·mL^−1^ for three reference solutions was prepared by dissolving and supplementing with 80% (*v*/*v*) ethanol to a constant volume of 25 mL.

For Chinese herb slices, 20.0 g of herb slices was placed into a decocting jar, and 250 mL of purified water was added to it, soaking for 30 min. Next, the mixture was decocted for 30 min on MSD-1-12 ceramic decocting equipment produced by Meisidi Craft Products Factory (Caozhou, China), and then the extracted liquid was poured out of the jar to obtain decoction I. Subsequently, another 250 mL of purified water was added to the decocting jar and the herb residue was decocted for 20 min, and the extracted liquid was poured out to obtain decoction II. Finally, decoctions I and II were put together to obtain the decoction of each herb. For the detection of choline analogs, 100 mL of the decoction was concentrated into a small amount of mixture under vacuum, which was then transferred to a 10 mL volumetric flask to make a constant volume with purified water. Then, 1000 μL of the suspension was transferred to a centrifuge tube with a pipette, diluting with equal volume of 80% ethanol (*v*/*v*), and centrifuged to obtain the supernatant. After filtering with a 0.45 μm microporous membrane, the sample solution for the decoction of each herb was obtained.

Fresh samples were cut into thin slices and then dried under 60 °C ambient temperature in a blast drying oven. All dried samples with the amount of 20 to 30 g were crushed into coarse powder and then passed through a 65-mesh sieve to obtain their corresponding powders. Then, 2.0 g of each powder sample was placed into a 50 mL Erlenmeyer flask with a stopper, to which 25 mL of 80% (*v*/*v*) ethanol was added. After sonicating twice for 10 min each time in a DK-410T water bath sonicator with a frequency of 40 kHz, the mixture was filtered, and the residue was washed twice with 80% (*v*/*v*) ethanol. The filtrate was concentrated under vacuum, and the residual suspension was transferred into a volumetric flask to prepare 2 mL of the sample solution with 80% (*v*/*v*) ethanol.

#### 4.2.3. TLC Analysis for Choline Analogs in Chinese Herbs

According to the general procedure of TLC analysis, the reference solutions of choline, *γ*-aminobutyric acid and synephrine were each sampled on a thin-layer plate, and then developed with solvent systems I (ethyl acetate–95% ethanol–formic acid) and II (*n*-butanol–glacial acetic acid–water), respectively. After drying, the thin-layer plates were visualized with Dragendorff, Wagner and improved Dragendorff reagents.

According to the optimized procedure, the TLC analyses for sample solutions from Chinese herbs *Zhishi*, *Zhiqiao*, *Qingpi* and *Chenpi* were performed for discovering probable choline analogs. After this, other herbs from the *Citrus* genus, including *Foushou* and *Xiangyuan*, and some samples of the leaves from the plants of the *Citrus* genus were also detected for possible choline analogs. Moreover, to further verify the efficiency of the method for detecting choline analogs, the TLC analyses for sample solutions from some other Chinese herbs, including *Huanglian*, *Juhua*, *Mahuang*, *Chuanxiong*, *Dafupi*, *Banxia*, *Duzhong*, *Kushen* and *Gancao*, were also determined. Among these herbs, it was reported that herbs *Chuanxiong* and *Huanglian* contain choline or its analogs.

### 4.3. Isolation and Identification of Choline Analogs in Chinese Herbs Zhishi

According to the sample solution process in Section 4.2.2, a solution of the *Zhishi* mixed sample from 10 different producing areas was prepared. Using preparative TLC, probable choline analogs in the Chinese herb *Zhishi* were isolated. Solvent I (ethyl acetate–95% ethanol–formic acid, 10:4:5) was used as the developing agent, and probable choline analogs were located by visualization of the slices of thin-layer plates immersed in the improved Dragendorff′s reagent.

Compounds **1** and **2** were identified based on the spectral analyses of their NMR and MS (Appendix A), combining their physicochemical analyses and the comparison of reported data [17,18]. The NMR data were recorded on a Bruker AV-600 or AV-400 MHz NMR spectrometer, and MeOH-*d*_4_ and D_2_O were used as the solvents dissolving compounds **1** and **2**, respectively. Compound **3** was identified based on the spectral analyses for its HRMS, comparing the TLC profiles of synephrine, compound **3** and their mixture. All HRMS data were obtained with ESI ion source (positive ion mode) from a TripleTOF 5600^+^ hybrid quadrupole time-of-flight mass spectrometer system (AB Sciex, Framingham, MA, USA).

### 4.4. Quantitative Analyses of Chlorine and Stachydrine with TLCS Analysis

#### 4.4.1. Procedure of TLCS Analysis

Reference and sample solutions were prepared according to the procedure described in Section 4.2.2, and all of them were filtered by a 0.45 μm microporous membrane before use. Certain volumes of sample solutions and reference solutions of choline chloride or stachydrine hydrochloride were sampled on an analytical and glass-based silica gel TLC plate (Qingdao Ocean Chemical Co., Ltd., Qingdao, China) placed on a YOKO-TD electric strip spotter (Whyoko New Technology Development Co., Ltd., Wuhan, China), and the plate was placed in a developing chamber. After being saturated with the solvent of ethyl acetate–95% ethanol–formic acid (10:4:5, *v*/*v*/*v*) for 15 min, the spots were developed for a span length of 60 mm. Subsequently, the plate was taken from the chamber and dried, and then was immersed in the improved Dragendorff reagent for 2 to 4 s. After taking them from the stained jar for 20 min, the TLCS analyses were performed with the reflection absorption method for 50 min on a KH-3000 Plus TLC Scanner (Shanghai Keze Biochemical Technology Co., Ltd., Shanghai, China), and the measured/reference wavelengths for choline and stachydrine were 568/820 nm and 574/820 nm, respectively. The contents of choline or stachydrine in Chinese herbs from the Citrus genus were calculated from the mean peak areas of three bands of choline or stachydrine from reference and sample solutions.

In the above procedure, the measured/reference wavelengths and the chromogenic stability were simultaneously determined from the 10 min interval determination in 70 min for the absorption curves of choline and stachydrine bands on the TLC plate after being visualized, and the sampling volume was 5.0 μL. Depending on the typical chromatographic profile of TLC analyses for the sample solutions of Chinese citrus herbs, the resolution between choline (or stachydrine) and its nearest strip was calculated.

#### 4.4.2. Methodology Validation

According to the general procedure of methodology validation, the limit of detection (LOD), the limit of quantitation (LOQ), precision, repeatability, linearity and range were evaluated using the reference standard of choline or stachydrine. The reproducibility was assessed using the powder of Zhishi. The recovery was tested using the powder of Zhishi with a known content of choline or stachydrine, and the reference standard of choline or stachydrine.

In detail, the LOD was defined as the lowest concentration of choline or stachydrine at which the signal-to-noise ratio was from 3 to 4, and the LOQ was defined as the lowest concentration of choline or stachydrine at which the signal-to-noise ratio was greater than or equal to 10 with a precision below 5%. According to the procedure of TLCS analysis in Section 4.4.1, six scans of a strip of choline or stachydrine on a plate were performed for evaluating the repeatability, using a choline or stachydrine solution (1.0 mg·mL^−1^), and the sampling volume was 10.0 μL. Simultaneously, intra- and inter-plate precision were assessed by detecting choline or stachydrine solutions, respectively, on a thin-layer plate with six replicates and on six thin-layer plates. The linear correlation was established using seven sampling volumes (three replicates for each) at a concentration of 1.0 mg·mL^−1^ for choline or stachydrine solution. The reproducibility was evaluated by the quantitative determination of choline or stachydrine in Zhishi powder (No. 20101001) with six replicates, and the sampling volume was 2.0 μL. The recovery was tested from 9 samples prepared by adding an amount of choline or stachydrine into the powders of Zhishi (No. 20101001) with a known content of choline or stachydrine, including three different amounts for choline or stachydrine with three replicates for each.

#### 4.4.3. Quantitative Analyses for Samples

According to the validated procedure of TLC analyses, the quantitative analyses of choline or stachydrine in the 39 purchased samples, including 10 of Zhishi, 10 of Qingpi, 10 of Chenpi and 9 of Zhiqiao, were performed with the external standard method in triplicate for each sample. Three spots were sampled on the plate for each sample or standard solution. The sampling volume of each spot for the sample solutions of Zhishi was 2.0 μL, and that for the sample solutions of Zhiqiao, Qingpi or Chenpi was 3.0 μL. Simultaneously, the sampling volumes were 1.0 and 3.0 μL for the reference solutions of choline and stachydrine, respectively. After being visualized, each lane on the plates was scanned to obtain the peak areas of choline or stachydrine in samples and corresponding standards, and then the contents of choline and stachydrine in various samples were calculated.

### 4.5. Quantitative Analyses of Synephrine with HPLC

#### 4.5.1. Procedure of HPLC Analysis

A 5.0 mg·mL^−1^ stock solution for synephrine was prepared by dissolving 10.0 mg of synephrine into 80% (*v*/*v*) ethanol, and transferred into a 2 mL volumetric flask which was further supplemented with 80% (*v*/*v*) ethanol to a constant volume. From this stock, a concentration of 0.5 mg·mL^−1^ for standard solution was prepared by 10-fold dilution.

The concentrations of synephrine in sample solutions were determined with the external standard method, referring to the procedure of the Chinese pharmacopoeia [1]. Briefly, the quantitative analyses of synephrine were performed using a Waters e2695 separation system consisting of a model 2998 ultraviolet detector (Milford, CT, USA), and the detection wavelength was set at 275 nm. A SinoChrom ODS2 (4.6 mm × 250 mm, 5.0 µm) (Elite, Dalian, China) was used as the chromatographic column, and the temperature was kept at 30 °C. A methanol and phosphate buffer with a ratio of 67:33 (*v*/*v*) was used as the mobile phase of the isocratic elution, and the flow rate was set at 1.0 mL/min. The phosphate buffer was prepared by dissolving 0.60 g of KH_2_PO_4_, 1.00 g of sodium dodecyl sulfonate and 1.0 mL of glacial acetic acid into 700 mL of water, which was then transferred into a 1000 mL volumetric flask and subsequently supplemented with the required amount of water to make a constant volume. Moreover, the injection volume for all sample solutions and standard solutions was 10.0 μL. Depending on the typical chromatographic profile of HPLC analyses for sample solutions of Chinese citrus herbs, the resolution between synephrine and its nearest chromatographic peak was automatically calculated by the HPLC system.

#### 4.5.2. Methodology Validation

Similar to the validation of TLC analysis, the LOD was defined as the lowest concentration of synephrine at which the signal-to-noise ratio was from 3 to 4, and the LOQ was defined as the lowest concentration of synephrine at which the signal-to-noise ratio was greater than or equal to 10 with precision below 5%. The linear correlation was established using seven concentrations (0.05, 0.10, 0.20, 0.40, 0.80, 1.60 and 3.20 mg/mL) of synephrine solution with three replicates for each. Six injections for a standard solution with a concentration of 0.50 mg·mL^−1^ were performed to evaluate the repeatability. Using a powder sample of Zhishi (No. 2010001), the stability of a sample solution in 24 h was evaluated, and the reproducibility was also assessed by the quantitative determination of synephrine in the powder sample with six replicates. The recovery was tested from nine samples prepared by adding an amount of synephrine into the powders of Zhishi (No. 20101001), including three different amounts of synephrine with three replicates for each.

#### 4.5.3. Quantitative Analyses for Samples

All sample solutions used for the quantitative analyses of chlorine and stachydrine in Section 4.4 were simultaneously used for the quantitative analyses of synephrine according to the valid procedure, and the synephrine concentration of the reference solution was 0.5 mg·mL^−1^.

### 4.6. Statistical Analysis for the Contents of Three Ingredients in These Four Chinese Citrus Herbs

All statistical analyses were performed using the Excel program of Microsoft Office 2016, and a bilateral *t*-test was used for the comparison of two groups. A paired *t*-test was selected when the number of data was the same (both *n* = 10), while two-sample equivariance was selected for the *t*-test when the number of data was different (*n* = 10 and 9). A *p* value less than 0.05 shows that the data difference between two groups is significant, and that less than 0.01 indicates that the data difference between two groups is very significant.

### 4.7. Comprehensive Analyses for Pharmacological Effects of Four Chinese Citrus Herbs

Studies published in the past 10 years on the bioactivities of synephrine, chlorine and stachydrine were unsystematically searched using the Google Scholar search engine and the databases Medline, CNKI and RSC, using the keywords “pharmacological” or “activity” or “review”, and “synephrine” or “chlorine” or “stachydrine”. Furthermore, the relevant references in the obtained literature were also checked. Based on the literature results and the contents of synephrine, chlorine and stachydrine in these four Chinese citrus herbs, the pharmacological effects of these compounds were comprehensively analyzed, together with their contributions to the pharmacological effects of the four Chinese citrus herbs. Some reasonable explanations for the differences in the pharmacological activities of these four Chinese herbs have been presented here.

## 5. Conclusions

To discover some ingredients with cholinergic activity and further clarify possible reasons for the complex functions presented by four Chinese citrus herbs, a simple and specific method was first established for quickly discovering possible choline analogs, and stachydrine and choline were discovered in these citrus herbs. Then, a TLCS method was first established for the quantitative analyses of stachydrine and choline, and the contents of both ingredients and synephrine in 39 herb samples were determined. The results showed that stachydrine and synephrine have commensurate contents in these herbs, and the contents of the two ingredients and choline in these herbs present similar decreasing trends with the delay of harvest time. However, the contents of synephrine decrease the fastest, while those of stachydrine decrease the slowest. Based on these findings, the pharmacological activities and pharmacokinetics reported for stachydrine and synephrine were compared, and the results indicated that stachydrine and synephrine can be considered as a pair of bioactive equilibrists in these citrus herbs, especially for their effects on the cardio-cerebrovascular system. Additionally, the results confirmed that stachydrine plays an important role in the pharmacological functions of these citrus herbs, especially in dual-directionally regulating the uterus, and in various beneficial effects on the cardio-cerebrovascular system, kidney and liver. Considering these findings, some more scientifical and reasonable interpretations would be concluded with the help of network pharmacology analyses performed for the main ingredients and the various pharmacological functions of these citrus herb decoctions, and which would indicate that the component differences in the formation, content, water solubility, extractability and pharmacokinetic characteristics can lead to the differences in the pharmacological functions of these citrus herbs. Furthermore, the quality markers of these herbs should be reevaluated based on this study, the results of which would present a good reference for the quality control of these herbs.

## Figures and Tables

**Figure 1 molecules-28-03813-f001:**
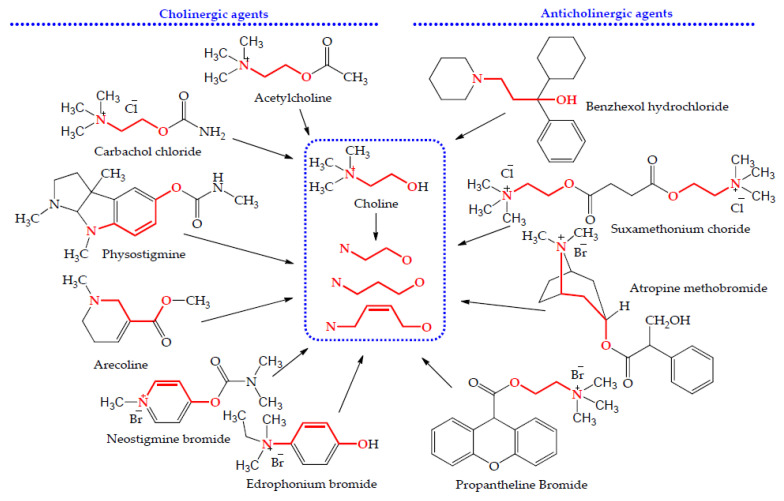
The extended structure–activity relationships of cholinergic and anti-cholinergic agents.

**Figure 2 molecules-28-03813-f002:**
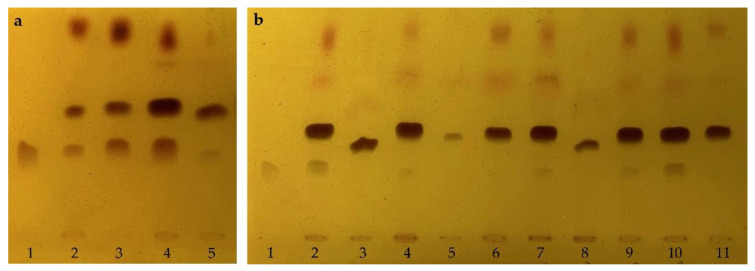
Detection of choline analogs in four Chinese herbs from the Citrus genus. (**a**) Chinese citrus herbs; 1, Choline; 2, Chenpi (No. 2010019); 3, Qingpi (No. 2010041); 4, Zhishi (No. 2010001); 5, Zhiqiao (No. 2010032). (**b**) Chinese herb Zhishi from different producing areas in China; 1, Choline; 2, No. 2008004 (Zizhong, Sichuan); 3, No. 2010002 (Shanggao, Jiangxi); 4, No. 2010003 (Danleng, Sichuan); 5, No. 2010004 (Baisha, Chongqing); 6, No. 2010005 (Anyue, Sichuan); 7, No. 2010006 (Ziyang, Sichuan); 8, No. 2010007 (Tonglian, Sichuan); 9, No. 2010008 (Jintang, Sichuan); 10, No. 2010009 (Jiangjin, Chongqing); 11, No. 2010010 (Lezhi, Sichuan).

**Figure 3 molecules-28-03813-f003:**
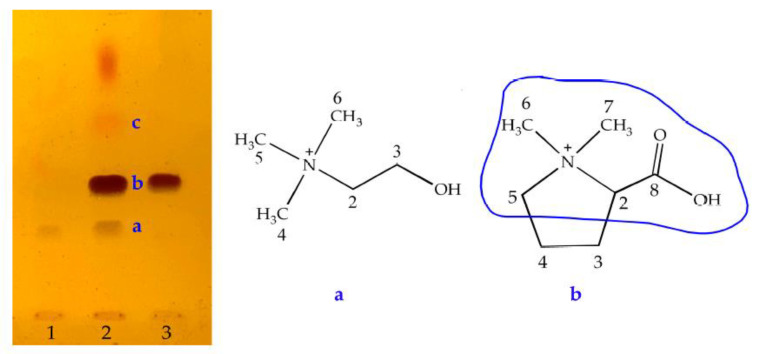
Chemical structure of probable choline analogs. **1**, choline; **2**, the mixed sample of Chinese herb *Zhishi*; **3**, stachydrine; spots (**a**–**c**) correspond to compounds **1**, **2** and **3**, and the chemical structures of choline analogs **1** and **2** are shown as (**a**,**b**), respectively, on the right side of the thin-layer plate.

**Figure 4 molecules-28-03813-f004:**
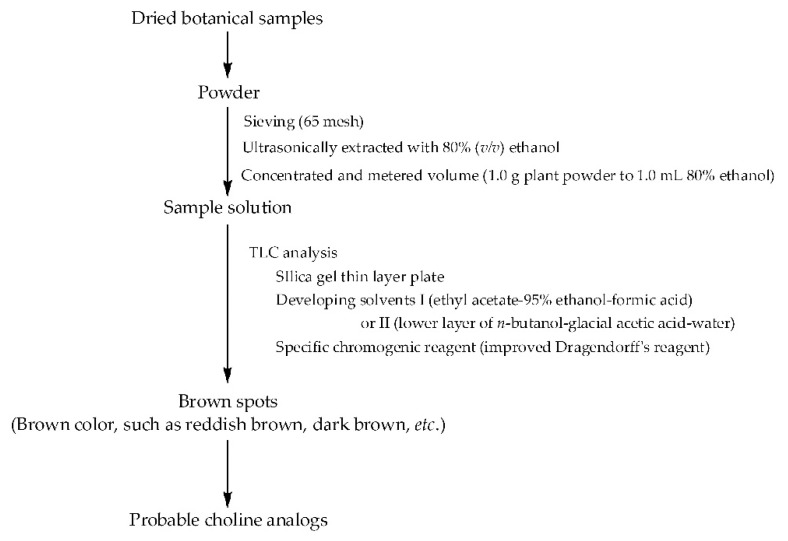
A simple procedure for quickly discovering choline analogs from botanical resources. The specific chromogenic reagent (improved Dragendorff’s reagent) was prepared by adding a mixture of bismuth nitrite (0.82 g), potassium iodide (11.06 g) and 50% (*v*/*v*) phosphoric acid (90 mL) into a 100 mL volumetric flask to make a constant volume using water.

**Table 1 molecules-28-03813-t001:** Contents of stachydrine, choline and synephrine in four Chinese herbs from Citrus genus ^a^.

Name of Chinese Herbs	Producing Area	Batch No.	Harvest Dates	Contents in Chinese Herb (mg·g^−1^)			Content Ranges (mg·g^−1^)		
				Stachydrine (SC)	Choline (CL)	Synephrine (SN)	Stachydrine (SC)	Choline (CL)	Synephrine (SN)
*Zhishi*	Zizhong, Sichuan	2010001	June 2020	3.10	0.87	11.03	2.94~8.53	0.00~0.87	3.56~21.07
	Shanggao, Jiangxi	2010002	June 2020	6.42	0.16	21.07			
	Danleng, Sichuan	2010003	June 2020	5.35	0.39	3.56			
	Baisha, Chongqing	2010004	June 2020	3.23	– ^b^	5.21			
	Anyue, Sichuan	2010005	June 2020	5.31	0.42	19.84			
	Ziyang, Sichuan	2010006	June 2020	6.46	0.47	20.10			
	Tongliang, Chongqing	2010007	June 2020	3.74	0.11	4.34			
	Jintang, Sichuan	2010008	June 2020	8.53	0.29	12.58			
	Jiasi, Chongqing	2010009	June 2020	2.94	0.21	9.64			
	Lezhi, Sichuan	2010010	June 2020	6.48	0.31	10.80			
*Zhiqiao*	Baisha, Chongqing	2010031	July 2020	1.43	0.10	2.42	1.43~5.13	0.00~0.21	0.00~2.42
	Tongnan, Chongqing	2010032	July 2020	2.51	0.21	1.82			
	Bazhong, Sichuan	2010033	July 2020	5.13	0.14	0.59			
	Zizhong, Sichuan	2010034	July 2020	3.92	0.20	1.25			
	Jiasi, Chongqing	2010035	July 2020	3.11	0.12	1.77			
	Zhangshu, Jiangxi	2010036	July 2020	2.31	0.19	1.73			
	Dazu, Chongqing	2010037	July 2020	2.66	0.19	1.63			
	Dazhu, Sichuan	2010038	July 2020	4.68	–	–			
	Ji’an, Jiangxi	2010039	July 2020	1.62	–	–			
*Qingpi*	Quzhou, Zhejiang	2010041	July 2020	2.35	0.34	5.16	0.99~3.51	0.21~0.60	4.39~7.19
	Zhangshu, Jiangxi	2010042	July 2020	2.63	0.32	6.18			
	Danleng, Sichuan	2010043	July 2020	2.22	0.47	6.11			
	Fengyuzhen, Sichuan	2010044	July 2020	1.93	0.32	6.37			
	Ziyang, Sichuan	2010045	July 2020	0.99	0.29	5.30			
	Huangshui, Sichuan	2010046	July 2020	1.15	0.21	6.29			
	Meishan, Sichaun	2010047	July 2020	3.51	0.47	5.08			
	Pingshan, Sichuan	2010048	July 2020	1.64	0.60	7.19			
	Ji’an, Jiangxi	2010049	July 2020	0.99	0.59	4.39			
	Shuangliu, Sichuan	2010050	July 2020	2.36	0.53	5.56			
*Chenpi*	Jintang, Sichuan	2010011	January 2020	0.96	–	1.98	0.86~3.26	0.00~0.20	1.86~3.80
	Yibin, Sichuan	2010012	January 2020	1.03	–	2.04			
	Nanchong, Sichuan	2010013	January 2020	1.92	0.16	2.71			
	Meishan, Sichuan	2010014	January 2020	2.93	–	1.86			
	Neijiang, Sichuan	2010015	January 2020	0.86	0.12	2.74			
	Yiyang, Hunan	2010016	January 2020	3.26	–	3.80			
	Anyue, Sichuan	2010017	January 2020	1.33	0.18	2.43			
	Lezhi, Sichuan	2010018	January 2020	1.16	0.09	2.16			
	Dazhou, Sichuan	2010019	January 2020	1.89	0.20	2.52			
	Xinhui, Guangzhou	2010020	January 2020	1.43	–	2.74			

^a^: These citrus herbs were commercially available from Chengdu Huichu Technology Co., Ltd., Chengdu, China; herbs Zhishi and Zhiqiao are the dried young and near-mature fruits, respectively, of *Citrus aurantium* L. or its cultivated varieties, and herbs Qingpi and Chenpi are the dried peel from young (or immature) and mature fruits, respectively, of Citrus reticulata Blanco or its cultivated varieties. ^b^: –, the content was lower than the LOD and was set as 0.00 for subsequent calculations.

**Table 2 molecules-28-03813-t002:** Statistical analyses for the contents of stachydrine, choline and synephrine in four citrus herbs ^a^.

Name of Chinese Herbs	Average Content ± SD (mg/g) ^b^			Sequencing of the Contents of SC, CL and SN ^c^	Sequencing of the Contents of SN, and SC Plus SC ^d^
	Stachydrine (SC)	Choline (CL)	Synephrine (SN)		
*Zhishi*	5.16 ± 1.87 **^##++^	0.32 ± 0.24 *^#^	11.82 ± 6.61 ^##+^	SN ^!!^ > SC > CL ^!!^	SN > (SC + CL) ^‡‡^
*Zhiqiao*	3.04 ± 1.29 ^#+^	0.13 ± 0.08 ^++^	1.25 ± 0.86 ^##++^	SC ^§^ > SN > CL ^§§^	(SC + CL) ^‡^ > SN
*Qingpi*	1.98 ± 0.81 *	0.41 ± 0.14 **^##^	5.76 ± 0.81 **^##^	SN ^!^ > SC > CL ^!!^	SN > (SC + CL) ^‡‡^
*Chenpi*	1.68 ± 0.83 *	0.08 ± 0.08 ^++^	2.50 ± 0.56 ^++^	SN ^!!^ > SC > CL ^!!^	SN > (SC + CL) ^‡^
Sequencing in herbs	*Zhishi* > *Zhiqiao* > *Qingpi* (*Chenpi*)	*Qingpi* (*Zhishi*) > *Zhiqiao* (*Chenpi*)	*Zhishi* > *Qingpi* > *Chenpi* > *Zhiqiao*		

^a^: The data before analysis are shown in Table 1. ^b^: *, ^+^ and ^#^ indicate that the differences are significant (*p* < 0.05) compared with Zhiqiao, Qingpi and Chenpi, respectively; **, ^++^ and ^##^ indicate that the differences are significant (*p* < 0.01) compared with Zhiqiao, Qingpi and Chenpi, respectively. ^c^: ^!^ and ^§^ indicate that the differences are significant (*p* < 0.05) compared with stachydrine (SC) and synephrine (SN), respectively; ^!!^ and ^§§^ indicate that the differences are significant (*p* < 0.01) compared with stachydrine (SC) and synephrine (SN), respectively. ^d^: ^‡^ and ^‡‡^ indicate that the differences are significant (*p* < 0.05) or remarkably significant (*p* < 0.01) compared with synephrine (SN).

**Table 3 molecules-28-03813-t003:** Main bioactivities of stachydrine (plus choline) and synephrine ^a^.

Effected Tissues, Organs or Systems	Pharmacological Effects	
	Synephrine	Stachydrine (Choline)
Eye	Exciting *α*_1_-adrenoreceptor and dilating the pupil [21].	/
cardio-cerebrovascular system	A partial agonist of *α*_1_-adrenoreceptor and an antagonist of *α*_2_-adrenoreceptor, and can weakly bind on *α*_1_- and *α*_2_-adrenoreceptors. The effects on *β*_1_- and *β*_2_-adrenoreceptors are very small and can be ignored [4,5,10,22,23,24]. (1) Constricting peripheral blood vessels including mesenteric artery, and raising blood pressure; (2) Complex responses of the coronary artery by the excitation of α_1_-adrenoceptor and TAARs [22]; (3) Constricting aorta directly by the excitation of *α*_1_-adrenoceptor and 5-HT1D [25], not by 5-HT1B and *β*-receptor [23]; (4) Cerebral vasoconstriction deduced from it acting on the *α*_1_-adrenoceptor.	Cardiovascular system protection [26]: (1) Accelerating blood circulation, increasing coronary and myocardial blood flow in **adrenaline-induced** myocardial ischemia [27,28]; (2) Relieving myocardial necrosis, lowering blood viscosity and vascular resistance, improving microcirculation [27,28]; (3) Slowing heart rate and decreasing cardiac output [27,28]; (4) Suppressing and ameliorating myocardial fibrosis [29,30]; (5) Ameliorating **isoproterenol-induced** cardiac hypertrophy and fibrosis [31]; (6) Inhibiting **norepinephrine-induced** cardiomyocyte hypertrophy [32,33,34]; (7) Rapid vascular relaxation mediated by the activation of endothelial nitric oxide synthase in vascular endothelial cells [35]; (8) Ameliorating **endothelial dysfunction induced by homocysteine** [36].
Blood	Increasing the level of platelet [37].	Inhibiting platelet aggregation and ameliorating platelet-mediated thrombo-inflammation [27,28,38];
Neuroprotective effects	/	(1) Protecting the neuronal injury [39]; (2) Inhibiting inflammatory reactions and improving pathological changes after cerebral ischemia [40]; (3) Inhibition of neuronal apoptosis, improvement of energy metabolism disorder, and microcirculation of brain [41].
Respiratory system	No bronchial constriction [42].	Antitussive effects by reducing citric acid-induced coughing [43].
Digestive system	A partial agonist of *α*_1_-adrenoreceptor and an antagonist of *α*_2_-adrenoreceptor. (1) Relaxing the intestinal smooth muscle and the intestine [3]; (2) A modest reduction in contractions for rabbit duodenum [42]; (3) Both of the above are also supported with it is an antagonist of *α*_2_-adrenoreceptor [5].	(1) Treating non-alcoholic fatty liver disease [26]; (2) Ameliorating carbon tetrachloride-induced hepatic fibrosis [44]; (3) For choline, maintaining the function and health of liver [45,46].
Uterus	Uterine contraction (pregnancy), deduced from the fact that synephrine is an agonist α_1_-adrenoreceptor [47].	Regulation of uterus effect (pregnancy and non-pregnancy) [27,48]: (1) Stimulation of uterine contraction [49,50]; (2) Inhibition of convulsive uterus [51]; (3) Reducing uterine bleeding [52].
Blood sugar	Inhibiting *α*_1_-adrenoreceptor and *α*-glycosidase, and presenting a hypoglycemic effect which can be also deduced from it being an antagonist of *α*_2_-adrenoreceptor [5,53,54].	Ameliorating and protecting high-glucose-induced endothelial cell senescence by upregulation of SIRT1 and downregulation of p16^INK4A^ [55].
Anti-inflammatory effect	/	(1) Inhibition of TXB2 and IL-10 secretion, and production of NO [56]; (2) Inhibition of NF-κB and AKT signal pathways [57]; (3) Improvement of cellular membrane permeability, and inhibition of inflammatory factors and lipid peroxidation [58].
Antidepressant activity	Anti-depressant activity by modulating noradrenergic neurotransmission and stimulating *α*_1_-adrenoceptor [59,60,61].	/
Anti-obesity	Weight loss, anti-obesity, and regulating fat metabolism, due to that synephrine is a partial agonist *β*_3_-adrenoreceptor, and can weakly bind on *β*_3_-adrenoreceptor [62], together with lipolytic and thermogenic effects [63].	/
Renal protection	/	(1) Reducing and ameliorating renal interstitial fibrosis [64]; (2) Ameliorating hydrogen peroxide-induced renal tubular epithelial cell injury [65]; (3) Protecting adenine-induced chronic renal failure [66]; (4) Inducing diuresis [27].
Pharmacokinetics	Pharmacokinetic characteristics [67,68,69]: (1) Oral ingestion absorption was fast, and the time to peak is approximately ranged from 1 to 2 h after administration; (2) The biological half-life is about 2 h; (3) The bioavailability is approximately 22%; (4) The metabolism is exerted predominantly in the liver, and it can be rapidly removed from the bloodstream by hepatic uptake; (5) Cannot cross the blood–brain barrier	Pharmacokinetic characteristics [70,71]: (1) Rapid absorption after oral administration (2) Fast and extensive distribution; (3) The biological half-life is about 4 h; (4) The time to peak is approximately 3 h after administration; (5) The bioavailability is above 90%; (6) Most excreted from urine.

^a^: /, no related reports obtained.

## Data Availability

Not applicable.

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
