# Peer review of "Stachydrine, a Bioactive Equilibrist for Synephrine, Identified from Four Citrus Chinese Herbs"

_molecules, 2023, doi:10.3390/molecules28093813_

Round 1

Reviewer 1 Report

Thank you so much for extensive work, nonetheless I have some concerns over the work as follows:

1. Why authors did not use the scientific name of the citrus genus?

2. How the decoctions are prepared is not clearly mentioned

3. The title of the study seems not befitting. Authors have identified compounds from four citrus genus but no bioactivity is assessed while the title focuses the bioactive compounds. Additionally, the way of manuscript flow looks as if its a review.

4. The abstract is quite a qualitative data, quantitation is missing. In fact, the results look like highly qualitative and do not reflect the title of the research. The conclusion of the work seems unwarranted.

5. The conclusion of the introduction section completely lacks the target of the study

6. Are all compounds in Figure 1 isolated from these herbs? If not, are they justified to be used as review of literature? I think its redundant while a textual approach of the compounds might be more legible

7.  How many solvent systems the authors have used in TLCthroughout the study? I see only one solvent system which does not confirm even the Rf values look similar. Because Rf values of several compounds at a single solvent system may be similar but Rf values of several compounds need to be verified in different solvent systems.
8. Similar issues are likely for HPLC, using one solvent system based on retention time may not ensure the reproducibility of the study

9. The NMR spectra and HPLC chromatograms are missing for all the compounds

10. Justification of the bioactivities using table 3 is not appreciable because the works are not at all done by the authors. Authors can confine their study within the isolation and identification of Stachydrine and synephrin analogs. 

11. As a research article, literature supplementation is unnecessarily extended

12. Conclusion section needs to be concluded through inclusion of the limitation of study and approach to overcome that. Future direction needs to be more focused

13.The collected samples should have identification numbers/accession numbers/ identifier's information is as well important. What about the reposition of the samples in the herbarium

14. Specify the Microsoft program and version (Excel is a program of Microsoft)

15. What were the laboratory conditions for method validation?

16. In many places, authors have used a "certain volume/amount" which iare vague. Not appropriate, at least needs to mention the concentration of the sample ready for TLC. important the TLC plat is not specified, analytical of preparative? Glass of Aluminium foil-based? If so, manufacturer?

17. The use of figure 5 is not warranted, it could be a graphical abstract part but not the outcome of this study

18. There are some other typo and formatting/syntax errors

Author Response

Dear Reviewer,

My co-authors and I are very grateful to you for your careful review, good comments, kind reminder and valuable suggestions. We have amended the manuscript according to the issues raised by you, and have pleasure to submit the revised version, together with the response to all points., for your consideration.

Please see the attachment, which includes the answers to your comments (Pages 1 to 15) and related comments (Pages 16 to 40) attached on the original manuscript.

Many thanks for your kind attention!

Yours sincerely,

Ganjun Yuan

Reviewer 2 Report

The article is interesting and generally well written. It needs some minor correction before it can be published. The most recurring problem is the presence of sentences too long, difficult to understand. Council to review the entire manuscript.

Here below there are my suggestions:

In the abstract, at lines 13-15 the authors reported as follows: “Although many pharmacological functions of these herb decoctions can be clarified from the bioactivities of identified ingredients, some of them remain confusing”. In my opinion, the purpose of the work is not well clarified from this sentence.

Lines 257, 259, 288, 290: change friut with “fruit”.

The sentence at lines 297-302 is too long and confused: “Moreover, depending on their descending speeds and degrees of these three components in the fruits and pericarps along with the growth time of these Citrus genus plants , it was further inferred that synephrine and stachydrine distributed more in exocarp than in mesocarp, and while there is possibly no obvious distribution difference of choline in exocarp and mesocarp, since herbs Zhishi (or Zhiqiao) and Qingpi (or Chenpi) are respectively derived from the fruits and the pericarps”.  Please, rewrite it . Furthermore, at line 297 probably the term more appropriate is “decreasing speeds”.

 Lines 895-899, the sentence is too long, please reformulate it: “To further clarify the reasons of the confusing functions of four citrus Chinese herbs, a simple and specific method for quickly discovering possible choline analogs was first established based on the extended structure-activity relationships of cholinergic and anti-cholinergic agents, and stachydrine and choline were discovered from these citrus herb  decoctions.“ Furthermore, I suggest to change the sentence “confusing function”, since it is not very incisive to emphasize the purpose of the work, but rather belittles it.  

The authors reported various times the pharmacological function of citrus plants are related to the presence of various compounds classes, including coumarins. In this regard, it is suggested to refer to the following article: “Dietary Intake of Coumarins and Furocoumarins through Citrus Beverages: A Detailed Estimation by a HPLC-MS/MS Method Combined with the Linear Retention Index System by Arigò et al., Foods, 2021,  10(7), 1533” https://doi.org/10.3390/foods10071533,  in which the authors determined the content of coumarins and furocumarines in citrus flavoured beverages to correlate it with dietary intake.

Author Response

Dear Reviewer,

My co-authors and I are very grateful to you for your careful review, good comments, kindly reminder and valuable suggestions. We have amended the manuscript according to the issues raised by you, and have pleasure to submit the revised version, together with the response to all points, for your consideration.

Many thanks for your kind attention!

Yours sincerely,

Ganjun Yuan

Here are our answers to your comments.

Point 1: The article is interesting and generally well written. It needs some minor correction before it can be published. The most recurring problem is the presence of sentences too long, difficult to understand. Council to review the entire manuscript.

Response 1: Thank you for your careful review, valuable comments and kindly reminder! We had carefully performed extensive revision throughout the manuscript including references, such as the spelling, formatting, syntax, linguistic edit and expression.

Point 2: In the abstract, at lines 13-15 the authors reported as follows: “Although many pharmacological functions of these herb decoctions can be clarified from the bioactivities of identified ingredients, some of them remain confusing”. In my opinion, the purpose of the work is not well clarified from this sentence.

Response 2: Thank you for your valuable comments and kindly reminder! We had already revised this sentence as “Many ingredients were already identified from these herbs, and their various bioactivities provide some interpretations for the pharmacological functions of these herbs. However, the complex functions of these herbs imply undisclosed cholinergic activity” for your consideration.

Point 3: Lines 257, 259, 288, 290: change friut with “fruit”.

Response 3: Thank you for your careful review! We had already revised them.

Point 4: The sentence at lines 297-302 is too long and confused: “Moreover, depending on their descending speeds and degrees of these three components in the fruits and pericarps along with the growth time of these Citrus genus plants , it was further inferred that synephrine and stachydrine distributed more in exocarp than in mesocarp, and while there is possibly no obvious distribution difference of choline in exocarp and mesocarp, since herbs Zhishi (or Zhiqiao) and Qingpi (or Chenpi) are respectively derived from the fruits and the pericarps”.  Please, rewrite it. Furthermore, at line 297 probably the term more appropriate is “decreasing speeds”.

Response 4: Thank you for your careful review and valuable suggestions! We had already revised the word “descending” as “decreasing”, and rewritten the sentence as follows for your consideration.

“Moreover, since herbs Zhishi (or Zhiqiao) and Qingpi (or Chenpi) are respectively derived from the fruits and the pericarps, that synephrine and stachydrine distribute more in exocarp than in mesocarp could be further inferred from their decreasing speeds and degrees along with the growth time of these Citrus genus plants. However, there is possibly no obvious distribution difference of choline in exocarp and mesocarp.”

Point 5: Lines 895-899, the sentence is too long, please reformulate it: “To further clarify the reasons of the confusing functions of four citrus Chinese herbs, a simple and specific method for quickly discovering possible choline analogs was first established based on the extended structure-activity relationships of cholinergic and anti-cholinergic agents, and stachydrine and choline were discovered from these citrus herb  decoctions.“ Furthermore, I suggest to change the sentence “confusing function”, since it is not very incisive to emphasize the purpose of the work, but rather belittles it. 

Response 5: Thank you for your careful review and valuable suggestions! We had already rewritten the sentence as follows for your consideration.

“To discover some ingredients with cholinergic activity and further clarify possible reasons of the complex functions presented by four citrus Chinese herbs, a simple and specific method was first established for quickly discovering possible choline analogs, and stachydrine and choline were discovered from these citrus herbs.”

Moreover, the terms “confusing function” had been revised as “complex function” or “convoluted pharmacological function” for your consideration.

Point 6: The authors reported various times the pharmacological function of citrus plants is related to the presence of various compounds classes, including coumarins. In this regard, it is suggested to refer to the following article: “Dietary Intake of Coumarins and Furocoumarins through Citrus Beverages: A Detailed Estimation by a HPLC-MS/MS Method Combined with the Linear Retention Index System by Arigò et al., Foods, 2021, 10(7), 1533” https://doi.org/10.3390/foods10071533, in which the authors determined the content of coumarins and furocumarines in citrus flavoured beverages to correlate it with dietary intake.

Response 6: Thank you for your help to improve our work and present a valuable reference that we really want but haven't discovered! We had carefully read the paper and cited it as Reference 88. Thank you very much!

Other revision:

Revision 1: Extensive revision

We carefully performed extensive revision throughout the manuscript including references, sun as linguistic edit, expression, the spelling, grammar, and format. Thank you very much!

Round 2

Reviewer 1 Report

Thank you so much for addressing the queries and improving the manuscript accordingly. However, I can not agree with the way how you collected and used the sample. You must have to follow the scientific approach. Collection------identification by authorized taxonomist---------------collecting accession number for future references----------------Of course the herbarium sheet should be prepared

Author Response

Dear Reviewer,

My co-authors and I are very grateful to you for your careful review, objective comments, kind reminder and valuable suggestions. We have amended the throughout the manuscript according to your suggestion, and have pleasure to submit the revised version, together with the responses to your points, for your consideration.

Many thanks for your kind attention!

Yours sincerely,

Ganjun Yuan

Here are our answers to your comments.

Thank you so much for addressing the queries and improving the manuscript accordingly.

    Thank you very much for your careful review, valuable suggestions, and enthusiastic helps to improve our work!

Point 1: However, I cannot agree with the way how you collected and used the sample. You must have to follow the scientific approach. Collection------identification by authorized taxonomist---------------collecting accession number for future references----------------Of course the herbarium sheet should be prepared.

    Response 1: Thank you for your enthusiastic help and kind consideration! You are right! For collecting fresh samples, we should follow the scientific methods as you said. At that time, we just considered that the collected fresh samples were only used for the supplementary detection of choline analogues and not for our main and other aims in this study. So, we did not follow the scientific approach, just identified by a senior agronomist Yao Nie, not by an authorized taxonomist, and not prepared the herbarium sheet. With the help and reminder of you, we will improve our sample collections strictly according to the scientific approach in the future researches.

Point 2: English language and style are fine/minor spell check required.

    Response 2: Thank you for your careful review! We had carefully performed extensive revision again throughout the manuscript including references, such as the spelling, formatting, syntax, linguistic edit and expression.
